# Long-Term Behavior of Cement Mortars Based on Municipal Solid Waste Slag and Natural Zeolite—A Comprehensive Physico-Mechanical, Structural and Chemical Assessment

**DOI:** 10.3390/ma15031001

**Published:** 2022-01-27

**Authors:** Marta Thomas, Małgorzata Osińska, Agnieszka Ślosarczyk

**Affiliations:** 1Faculty of Civil and Transport Engineering, Institute of Building Engineering, Poznan University of Technology, 60-965 Poznań, Poland; marta.thomas@put.poznan.pl; 2Faculty of Chemical Technology, Institute of Chemistry and Technical Electrochemistry, Poznan University of Technology, 60-965 Poznań, Poland; malgorzata.osinska@put.poznan.pl

**Keywords:** municipal solid waste slag, cementitious composite, natural zeolite, metal leachability, mechanical parameters, long term behavior

## Abstract

Limitations in natural aggregate resources and the continuous increase in the demand for concrete as a building material, as well as the increase in the production of waste and the problem with its storage were the reasons for attempts to replace the sand fraction in cement matrices with a corresponding slag fraction. Municipal solid waste incineration (MSWI) slag, which is a product of waste incineration, can be used as an aggregate. This extends its service life and reduces landfill waste. Therefore, three types of cement mortars with different aggregate composition were prepared. In addition, to increase the durability of the cement matrix and the degree of immobilization of harmful heavy metals and salts present in the slag, a natural zeolite with pozzolanic properties was used. A set of tests was carried out on fresh mortar and hardened mortar, including strength tests after 7, 28 and 360 days. What is more, chemical tests were undertaken, including the content of chlorides and sulfates, leaching using the TCLP method and oxide composition. The conducted tests revealed that all mortars had similar strength properties and demonstrated the effectiveness of immobilizing harmful substances contained in the municipal solid waste incineration (MSWI) slag by cementing.

## 1. Introduction

The Waste Framework Directive was adopted by the European Parliament and Council in 2008 to protect the environment and human health. This Directive introduced the concepts and targets of waste management for European Union member states. It also defined the waste management hierarchy: prevention, reduction of source and reuse, composting and recycling, recovery of energy, treatment and disposal [1]. One of the treatment methods is an incineration process. This can reduce the mass of waste by over 70% and the volume by up to 90%. The main effects of applying this technology include processing of non-usable residual waste in such a way to neutralize (inertize) them and use them as a source of energy. Hence, during incineration, the energy is recovered and transformed to electric and heat energy. This is possible by converting the pressure, temperature and steam from the exhaust gases from waste incineration in the furnace into energy [2]. In addition, this method makes it possible to convert residues into useful raw materials [3]. Products of municipal solid waste incineration (MSWI) are as follows: fly ash (FA), bottom ash (BA) and slag [4]. Those combustion products can be reused in the following applications: for cement production—as a replacement for cement binder or fine and coarse aggregate (slag) in concrete; in ceramics production—by vitrafication in glass and glass-ceramics production; in general construction—as road pavement or in embankments [5]. By processing municipal waste and reusing the resulting by-products, landfilling of this waste can be significantly reduced. According to Eurostat, the amount of waste incinerated has increased by 100% in the last 25 years. Today, in Europe, about 60% of all waste is incinerated and more than 67% of all materials are recycled [6]. In contrast, in some Asian countries, including Japan, about 80% of waste is incinerated, recycled and reused. Of note: in the People’s Republic of China, over 80% of all waste still ends up in landfills [7].

Municipal solid waste incineration products, because of the high content of heavy metals and dioxins, are treated as hazardous materials. However, according to one manufacturer and research studies, seasoning the slag in the air for 1 to 6 months increases the resistance to leaching. This is due to the hydration process occurring in the grains of the slag under the influence of humidity [8,9,10,11]. MSWI products should be therefore immobilized [12].

The most popular immobilization processes are: geopolymerization, bituminization, vitrafication and cementation. Regarding the last, cementation is very effective way of attenuating heavy metals, and harmful substances retention can be up to 90% [13]. The C-S-H phase formed in the paste allows immobilization by cementing. Its large specific area, ability to attach other compounds to its structure and low permeability, allow the containment of hazardous substances. Cement matrices should have good physical and mechanical properties to obtain better durability of the elements. Properties of municipal solid waste incineration slag are similar to natural aggregates, and hence, can replace natural aggregate partially or completely. Using MSWI slag (useless waste) instead of natural and non-renewable aggregates is environmentally friendly and complies with waste management hierarchy principles.

One of the problems of using MSWI slag is that the composition of this raw material is unstable because it depends on the morphological composition of waste. Therefore, the structure of the slag can vary due to the economic or industrial region served by the specific incinerator, as well as due to the season of the year. This diversity also influences the content of toxic chemicals in the slag [14]. Hence, there is a need to verify local incinerator products as a substitute for cements or/and aggregate before its application in local mortars and concretes, especially in longer periods of time. MSWI slag, like recycled concrete aggregate, is characterized by higher porosity and thus higher water absorption, as compared to natural aggregate [15,16].

One way to counteract the mentioned technological problems when producing mortars or concretes incorporating slag from municipal waste incinerators is to thicken and strengthen the microstructure of the cement binder by using pozzolanic additives. An example of a pozzolanic additive is the zeolite used in this study. Zeolites are natural porous volcanic tuffs that are characterized by having high adsorptive capacity, low specific mass, as well as high specific surface. The excellent mechanical properties of zeolites have been confirmed in many experimental studies [17,18,19,20]. The mixture of cement and zeolite allows obtaining high strength concrete with greater compressive strength than in case of conventional Portland cement [19]. In addition, the characteristic reticular structure of zeolite allows it to function as an ion exchanger and a selective adsorbent. Both adsorption and ion exchange depend on the charges, as well as their dimensions [20]. Therefore, this research is focused on the stabilization/solidification method of immobilization of MSWI slag by cementation. Moreover, an important novelty of the research presented in this paper is the use of natural zeolite, which, due to its pozzolanic properties and ability to adsorb harmful substances, is additionally assumed to strengthen the microstructure of the cement matrix and improve the immobilization of heavy metals, chloride and sulphates substances, as well as to enhance the durability of the produced mortars.

In this study, physical and mechanical parameters of mortars utilizing MSWI slag were compared with mortars made from sand and a mixture of sand and MSWI slag. For this purpose, the finest slag fraction of 0–4 mm was selected. Furthermore, in contrast to previous studies conducted on the use of slag from waste incineration plants, in the present study, the composition of the designed mortars took into account the water absorbability of the slag. Such an approach ensured very good mixing of the components and obtaining good physical and mechanical mortar parameters. An important part of the study was to analyze mortar behavior over longer periods of time—one year after the samples were made. During the conducted research, matrices strength, heavy metals leachability, chloride and sulphates content for mortars with and without zeolite content were tested.

## 2. Materials and Methods

### 2.1. Cement, Zeolite and Slag Characteristics

Table 1 and Figure 1 show the oxide and phase compositions for cement, municipal slag and zeolite, respectively. The cement used is Portland cement, with a cement clinker content of 95%. The cement is overwhelmingly composed of calcium oxide, silicon and aluminum. Additionally, oxides of iron, sodium, sulfur manganese and strontium were isolated. In the phase composition, mainly tricalcium silicate compounds, calcium aluminates and carbonates, as well as calcium alumino-ferrite and calcium sulfate were identified. The oxide composition of slag from municipal waste incineration plants is much richer compared to that of cement. Additionally, the oxides also identified for cement, and the presence of copper, titanium, zinc and chromium oxides was detected in the slag. Despite the varied composition, the main slag phases are calcium sulfates, calcium carbonates, calcium oxide and silicon dioxide. In contrast, the oxide composition of the zeolite is typical of natural clinoptilolite material, as confirmed by X-ray-structural analysis. The high silicon dioxide content, at 60%, indicates the potential pozzolanic properties of the zeolite.

Figure 2 shows the thermogravimetric curves for cement, slag and zeolite. The curves indicate that the cement exhibits the highest temperature stability, with a weight loss of only 3% over the entire temperature range studied. The slag also shows good temperature resistance, with a 5% weight loss recorded at around 1000 °C. The zeolite is the least stable, showing a weight loss of about 10% in the studied temperature range. Considering the changes in the DTG curve for cement and slag, one can clearly see the same trends in the decomposition of the compound corresponding successively, at temperatures up to 200 °C to the evaporation of water and removal of constitutional water, then at about 400 °C to the decomposition of silicate phases and at about 500–600 °C to the decomposition of portlandite. In the case of slag, additionally, a peak above 800 °C is clearly visible in the DTG curve corresponding to the decomposition of gypsum, one of the main phases recorded in the X-ray-structural analysis for slag from municipal waste incinerators [17]. The similar behavior in the DTG curves is due to the like oxide composition of the cement and slag, with a clear indication of the lower stability of these phases for the slag. The DTG curve recorded for zeolite shows one broad peak at temperatures up to 400 °C corresponding to clinoptilolite decomposition, which has been confirmed by other investigators [18].

The graining of slag is similar to the graining of sand (Figure 3). The only difference is that 85% of the sand grain size is contained within the 0.25 to 1 mm fraction, while 99% of the slag grain size being in the fraction from 0.5 to 4 mm.

The bulk density of aggregate used for the tests is given in Table 2. The bulk density of slag is smaller than the bulk density of sand. Accordingly, the slag loose bulk density is only 66% of the sand loose bulk density, while slag tapped bulk density is around 70% of sand tapped bulk density.

### 2.2. Cement Mortar Preparation

CEM I 42.5 R Górażdże is a hydraulic binder that is manufactured by co-grinding Portland clinker (main component) and sulphate raw material (setting time regulator). This kind of cement is suitable for the production of concrete mixtures containing additions of fly ash and ground blast furnace slag, allowing the optimum use of their pozzolanic and hydraulic properties. The 42.5 designation describes successively: cement strength class (compression strength after 28 days equal to 42.5 MPa) and strength growth rate determined after 2 or 7 days (R—cement with high early strength) [21]. The processed slag consists of silica, metal oxides and a small amount of unburned combustibles and water. The material was supplied by ITPOK (Installation for Thermal Transformation of Municipal Waste in Poznań, Poland). The slag of the 0–10 mm fraction was dried using a gas burner and sieved to separate the 0–4 mm fraction. The sand used for mortars is fine aggregate. There are two varieties of sand normally employed: 1—with grains up to 2 mm in diameter and 2—with grains of diameter up to 1 mm. Sand of the first variety is used for cement mortars (KWARCMIX, Tomaszów Mazowiecki, Poland).

### 2.3. Determination of Consistency of Fresh Mortar

Determination of fresh mortar consistency was according to the standard [22]. The glass disc was wiped by a wet cloth, and then lubricated by the application of mineral oil. The mold was placed centrally on the discs of the flow table (larger diameter facing downwards) and the filing adapter was placed upon the mold. The mortar was introduced in two layers, each layer being compacted by at least 10 short strokes of the tamper. The adapter was then removed, and the excess mortar was wiped away. Subsequently, the free area of the disc was wiped clean. After approximately 15 s, the mold was slowly raised vertically and removed. The flow table was then jolted 15 times by rotation of the crank at constant frequency of approximately one per second. Immediately after the last rotation, the diameter of the mortar was measured in two directions at right angles to one another. The mean value of diameter was then calculated.

### 2.4. Determination of Bulk Density

Bulk density was determined according to accepted standards [23]. The test portion was first dried by heating at a temperature of 110 ± 5 °C to constant mass and was then allowed to cool. An approved empty cylinder was weighed, and its mass reported as m_1_. The container was then filled with aggregate until it overfilled. The excess material was subsequently removed by rolling the tamping tool across the top surface. The container with the sample was then weighed and its mass reported as m_2_. The loose bulk density was then derived. Following the above, the container with the sample was placed upon the vibrating desk. The flange was placed upon the container and filled with the aggregate to about two thirds full. The sample was then tapped. The flange was removed, and the excess material was stripped away by rolling the tamping tod across the surface to obtain the level of the top of cylinder. The container with the tapped sample was subsequently weighed and its mass has been reported as m_3_. The tapped bulk density was then calculated.

### 2.5. Determination Flexural and Compressive Strength

Flexural and compressive strength were assessed according to accepted practices [24]. In order to ascertain the mechanical properties of the samples, molds of standard dimensions were used (40 mm wide, 40 mm high, 160 mm long). The molds were covered with antiadhesive liquid and then filled with the cement samples. The cement matrix samples were demolded after 24 h, and left for 7, 28, 56 and 360 days in a water environment so they can obtain adequate compression and bending strength. In order to receive reliable results, six samples of each recipe were prepared. Bending strength was determined by utilizing a hydraulic press. Each sample was weighed first. The samples were processed to destruction at rates of 2 mm/min, with initial value of 50 N rising to a max of 3000 N/min under stress control, the debris was retained separately for the subsequent chemical testing. Compressive strength was ascertained by means of a hydraulic press. Each sample was weighed first. The samples were processed to destruction at rates of 144 kN/min. the debris was retained separately for the subsequent chemical testing.

### 2.6. Determination of Total Sulfate Content

Total sulfur content was derived through acid digestion (Reference method) according to accepted standards [25]. Approximately 20 g of sample was crushed and sieved to pass the 125 µm sieve. The test portion was then weighed. Afterwards, distilled water and hydrogen peroxide were added. The sample was then warmed. After the dissolution, 20 mL of hydrochloric acid was added, and the specimen was heated. Filter paper pulp was then introduced. Subsequently, ammonium hydroxide was added to make the solution alkaline, and this was then filtered. The filtrates were reserved, and were transferred into a beaker and dissolved in concentrated hydrochloric acid and hot water. The procedure was repeated (boil, precipitate, filter and wash) with precipitate rejected. The combined filtrates and washings were then acidified with concentrated hydrochloric acid and brought to boiling. Next, heated barium chloride solution was added. The solution was then returned to boiling. Afterwards, the solution was left in a warm place overnight. The next day, the specimen was filtered and washed with warm demineralized water through fine filter paper. The filter paper together with the sediment was moved into a roasted and weighed crucible. The crucible and filter paper were dried and then the paper slowly burned in an oxidizing atmosphere. After cooling to room temperature, the crucible was weighed.

The total sulfate content was calculated from the following formula:SO_4_ = m_7_/m_6_ · 41.16,(1)
where: m_7_—the mass of precipitate in grams; m_6_—the mass of the test portion in grams.

### 2.7. Determination of Chloride Content

Chloride content was determined by applying Volhard’s method according to the standard [26]. Approximately 1 g of crushed material was taken as the test portion. The weighed sample was placed in a beaker and then wetted. Next, nitric acid and hot water were added, the mixture was subsequently heated to boiling and stirring continuously. The mixture was then filtered immediately by using medium filter paper. The stirrer, beaker and the residue on the filter were washed. Silver nitrate solution was then infused into the test solution. Afterwards, the mixture was stirred vigorously to precipitate the chloride. Following the above, the indicator solution was added and was titrated with the ammonium thiocyanate solution one drop at a time, while continually agitating the solution until the faint reddish-brown coloration no longer appeared. The volume V_1_ of solution used in the titration was recorded. Assessment of the no-concrete test portion was carried out using the same procedure, V_2_ was obtained in the blank titration.

The chloride content was calculated as a percent of chloride ion by mass of sample using the following formula:CC = 3.545·f·V_2_ − V_1_)/m,(2)
where: V_1_—volume of ammonium thiocyanate solution used in the titration (mL); V_2_—volume of ammonium thiocyanate solution used in the blank titration (mL); m—mass of the concrete sample (g); f—molarity of silver nitrate solution

### 2.8. Toxicity Characteristic Leaching Procedure (TCLP)

The TCLP testing was carried out using the samples that were previously destroyed during the mechanical studies. On the basis of the acid meter readings, 4.93 pH extraction fluid for pure slag was used and 2.88 pH for cement mortars. The leaching procedure to assess toxicity is schematically presented in Figure 4 and was undertaken according to standard [27].

Determination of metal concentrations was accomplished by applying the Atomic Absorption Spectrometry (AAS) method. As this is a relative method, to determine the concentrations of individual elements, standard curves are used. In order to create these, stock solutions containing 1 g/L of the studied ions were prepared (Cu, Pb, Ni, Zn, Fe and Cd). The working solutions of various concentrations were then obtained by appropriate dilution to 1, 2, 3, 5, 10, 15, 20 and 30 mg/L. After testing the absorbance of individual solutions, standard curves were created. Using the aforementioned standard curves, approximate values were obtained for the concentration of specific metals in a given solution.

### 2.9. Microstructure Research

In order to establish the mechanism of contact zone formation, studies were run utilizing the scanning microscope: TESCAN3VEGA. To observe the microstructure of the cement mortars, the samples were crumbled in order to create pieces of approximately 1 cm^2^ area with flat surfaces. All the samples were 28 days old and dried for 24 h at a temperature of 105 °C.

### 2.10. XRF/XRD Analysis

In this analysis, measurement conditions were: 25 kV accelerating voltage; 3 uA current; measurement time 200 s. The measurements were conducted in a flow of nitrogen and the samples were centrifuged during the measurement. The XRD analyses were performed using the Bruker AXS D8 Advance diffractometer (Bruker, Billerica, MA, USA) with Johansson monochromator (λCu Kα1 = 1.5406 Å).

### 2.11. TG/DTG Analysis

The TGA/DTG tests was determined using the Jupiter STA 449F3 apparatus (Netzsch GmbH, Waldkraiburg, Germany). The heating rate was equal to 10 °C/min from the ambient temperature up to 1000 °C for all the temperature plateaus under a nitrogen atmosphere.

## 3. Results

### 3.1. The Influence of Additives and Slag on Mortar Plasticity

Figure 5 shows the influence of additives and the slag content on the consistency of the fresh mortars.

Mortars with higher slag content are characterized by lower plasticity due to higher porosity, low density and greater water demand of the slag as aggregate. The addition of zeolite also reduces the plasticity of the fresh mortars compared to those without zeolite.

### 3.2. The Influence of Additives and Slag on Flexural Strength of Cement Mortars

The mortars were tested for bending strength after 7, 28 and 360 days. Flexural strength results are presented in Table 3. The sample with sand has the highest strength after 7 days, the strength of sample with slag and zeolite is comparable. The samples with a mix of aggregates have low initial bending strength. Compared to the samples tested after 7 days, the flexural strength of all samples increased after 28 days. The greatest increase in endurance occurred between 7 and 28 days and was recorded in samples with sand and zeolite and amounted to 33%, while the increase of samples with mixed aggregate was 30%. After 360 days, the bending strengths of all types of samples were found to be greater than after 28 days. The greatest increase in strength was recorded in samples with sand—18%, and in samples of mixed aggregate—20%. Samples based on sand, both with and without zeolite showed the highest bending strength after 360 days, followed samples based on slag alone. The worst bending strength results were shown by samples based on mixed aggregate. It is worth noting that most samples with zeolite exhibited higher flexural strength than did samples without zealite, but with the same proportion of aggregates.

### 3.3. Compressive Strength of Cement Mortars

Compressive strength of mortar was tested 3 times for each part, after 7 days, 28 days and 360 days. The results of compressive tests of samples without zeolite are shown in Figure 6, while results of samples with zeolite are shown in Figure 7.

It can be noticed that the highest initial (after 7 days) compressive strength was indicated in samples with sand only (28 MPa), while samples with slag had very similar endurance. The compressive strength of all types of samples was comparable after 28 days. After one year, a significant increase in compressive strength was noted in all samples. The highest compressive strength was found for samples based on slag, slightly lower results were indicated for samples based on sand alone, and the lowest results were achieved for samples containing a mixture of aggregates.

The samples with zeolite show generally higher compressive strength than did corresponding samples without zeolite. The sample with the addition of zeolite based on slag alone showed a significantly higher initial strength than the sample without zeolite. Its strength was the highest initial strength. Strength after 28 days was comparable for all three types of samples. A significant increase in compressive strength was noted in all samples after 360 days. Samples based on slag only showed the highest compressive strength after one year, samples based on mixed aggregate showed lower, and the lowest compressive strength was seen in samples based on sand alone.

### 3.4. Leachebility of Chemical Compounds

It was important to ascertain if the solidified slag in cement mortar is safe for the environment, and whether heavy metals are bound with cement firmly enough, and, for example, atmospheric conditions will not lead to gradual leaching of metals to the environment. For this purpose, leachability tests were carried out. The results of leaching trials performed for pure slag are shown in Table 4. In eluates from leaching by HCl and HNO_3_, all contained metals were present. However, eluates from leaching tests by means of the TCLP method did not contain lead, copper and nickel. In the leachate, the highest concentration of iron was detected.

To investigate the effect of time on the leaching of metals from cement mortars, leaching tests were carried out after 90 and 360 days of aging. Leachates obtained for every mortar sample containing slag were, after 90 days of setting, free of copper, lead, nickel, iron and cadmium. In these leachates, only zinc was detected, but its concentration did not exceed 0.5 mg/L (Table 5). Zinc was also present in elutes from the samples leached after 360 days. The results of leaching trials performed after 360 days for sample 1 and 2 also showed the presence of lead and iron. However, their concentration in leachate was less than permissible according to a TCLP test [27]. Moreover, the concentration permissible for water eluates was not exceeded for any of the tested samples and for any of the metals. The performed examination evidenced that slag present in mortar is not dangerous for the environment due to effective immobilization of the metals within the cement matrix.

Chloride and sulfate content was determined for slag and all samples after 90 and 360 days. The results of the determinations are presented in Table 6.

Samples with a higher percentage of slag show a higher content of sulfates and chlorides. After 90 days, slag-based samples without zeolite show lower sulfate content than the corresponding samples with zeolite. Furthermore, samples containing the slag without the addition of zeolite show a slight increase in the content of sulfates, while samples based on slag with the addition of zeolite show a significant decrease in content of sulfates between 90 and 360 days. Different results were obtained for samples based only on sand. Here, a higher sulfate content was noted after 90 days in the sample without zeolite than in the sample with zeolite. Moreover, between 90 and 360 days, a decrease in the content of sulfates was recorded both in the samples with and without zeolite.

Our work indicates that the chloride content of mortars increases as the amount of slag in the sample increases. The lowest chloride contents were recorded for samples made using only sand. Overall, though, the addition of zeolite slightly reduces the chloride content in the samples after 90 days. Indeed, chloride content in the sample tested after 360 days is much lower than the chloride content after 90 days. This is due to a significant thickening of the cement matrix microstructure resulting from cement hydration and pozzolanic reaction between zeolite and cement binder. In this reaction, silicon dioxide present in zeolite reacts with calcium hydroxide formed during hydration of silicate phases of the cement binder and an additional C–S–H phase is formed. This results in an increase in compressive strength and a higher compaction of the microstructure, leading to a reduction in the porosity of the cement mortar and lower leachability of chemicals. Similar relationships were also obtained by other researchers for mortars with natural zeolite [28].

### 3.5. Microstructural Characterization of Cement Composites

Figure 8 shows the scanning electron microscopy images of cement mortars made with pure slag and pure sand without and with zeolite addition, taking into account the evaluation of mortar microstructure and contact zone at the aggregate cement paste interface. Mortars of pure sand show a very well compacted cement matrix microstructure that is characteristic for a binder based on pure cement clinker (Figure 8a). In addition, the uniformity of the material without visible cracks and pores and good adhesion of the cement paste to the aggregate draws attention. Nevertheless, cracks can be observed at the cement paste-aggregate interface, resulting from the weakening of this zone by portlandite precipitated during cement hydration.

The situation is completely different for mortar with zeolite (Figure 8a’). The microstructure of the cement matrix is visibly thickened and the adhesion of the cement paste to the aggregate is improved. There is no microcracking at the cement paste-aggregate interface, this is due to the pozzolanic reaction between the zeolite and calcium hydroxide with the cement binder. This reaction results in the formation of the C–S–H phase and sealing of the mortar microstructure. Similar relationships were obtained for mortars incorporating slag from municipal waste incinerators (Figure 8b,b’). Noteworthy is the variety of aggregate in the form of slag. The photos clearly show fragments of aggregate characterized by increased irregularity and porosity. This contributes to the increased adhesion of the aggregate to the cement paste, which can be seen in the picture. Nevertheless, the aggregate from the municipal waste incinerator contains fragments with glassy surface and poor adhesion to cement paste, which was also confirmed by scanning microscope analysis (Figure 8c,c’).

## 4. Conclusions

Based on the conducted research, the following conclusions can be drawn:MSWI slag has similar graining to sand, while the bulk loose and tapped density of MSWI slag is lower than appropriate sand densities. This is due to the greater porosity of the slag and results in greater water demand and worse workability of the mortar.The flexural strength of mortars based on MSWI slag alone and sand alone is comparable and increases with time. The addition of zeolite improves the flexural strength of cement mortars by about 4.2% and 6.4%, respectively, for mortars with 100% slag and 100% sand content.The compressive strength of mortars based on slag alone and sand alone is comparable and increases with time. The addition of zeolite improves the compressive strength of cement mortars by about 6% and 1.7%, respectively, for mortars with 100% slag and 100% sand content. The compressive strength of mortar with mixed aggregate is lower than that of mortars based on slag alone by about 6 and 12%, respectively, for mortars with and without zeolite.After 90 days of curing, in the leachates, only zinc was detected, but its concentration did not exceed 0.5 mg/L. Zinc was also present in elutes from the samples leached after 360 days. The results of leaching trials performed after 360 days for sample 1 and 2 also showed the presence of lead and iron. However, the acceptable concentration was not exceeded for any of the tested samples and for any of the metals.The higher the slag content, the higher the sulfate content. Sulfate content of slag-based mortars increases with time, while adding zeolite to the same recipes reduces the sulfate content in the mortar by about 22%.The higher the slag content, the higher the chloride content. The chloride content decreases with time in all the tested mortars.A positive effect of zeolite action was also observed. Significantly lower levels of metal leaching and lower levels of sulfate and chloride were observed in all samples that incorporated zeolite. This is due to its pozzolanic properties. During the hydration of the cement binder, zeolite reacts with calcium hydroxide to form a C–S–H phase, which thickens the microstructure of the mortar, thus reducing the leachability of the substance.The conducted research indicates that it is possible to replace the natural aggregate of the sand fraction with MSWI slag of the same fraction in cement mortars. This action does not significantly affect the deterioration of the mechanical properties of the hardened mortar. Moreover, the obtained results establish that the stabilization of hazardous substances contained in MSWI slag is effective. However, in order to widely use cement mortars based on MSWI slag, subsequent tests, including durability tests of such mortars, are necessary.

## Figures and Tables

**Figure 1 materials-15-01001-f001:**
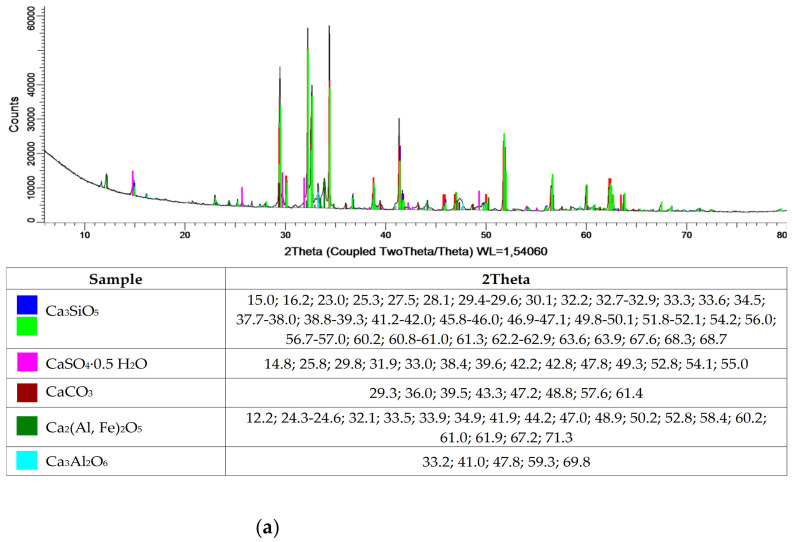
XRD characterization of cement (**a**), slag (**b**) and zeolite (**c**).

**Figure 2 materials-15-01001-f002:**
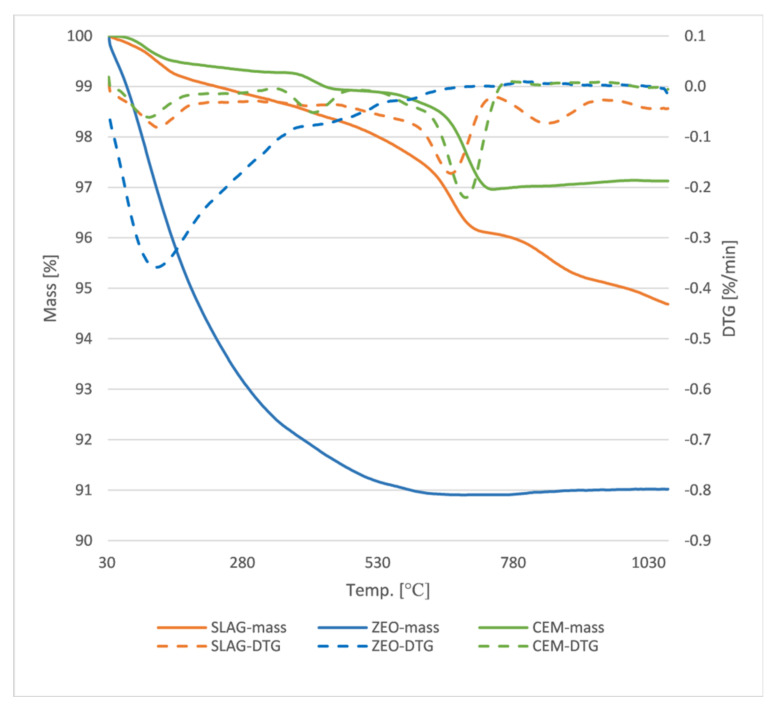
TG/DTG characterization of slag, zeolite and cement.

**Figure 3 materials-15-01001-f003:**
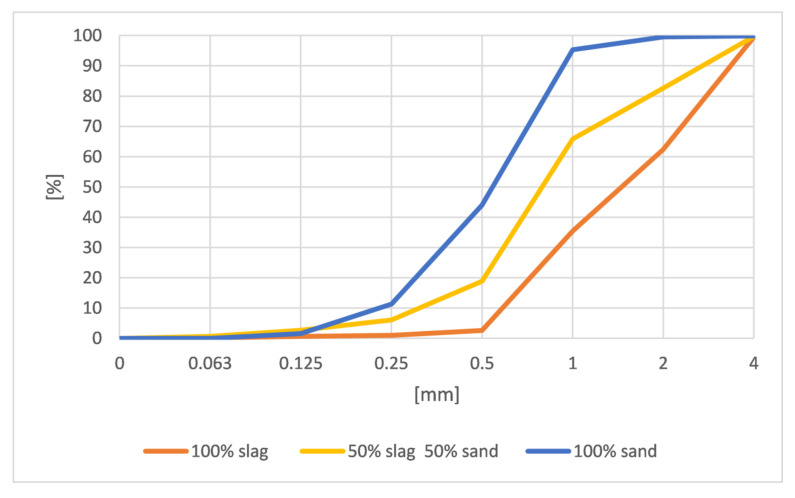
The grading curves of all types of aggregate mixtures.

**Figure 4 materials-15-01001-f004:**
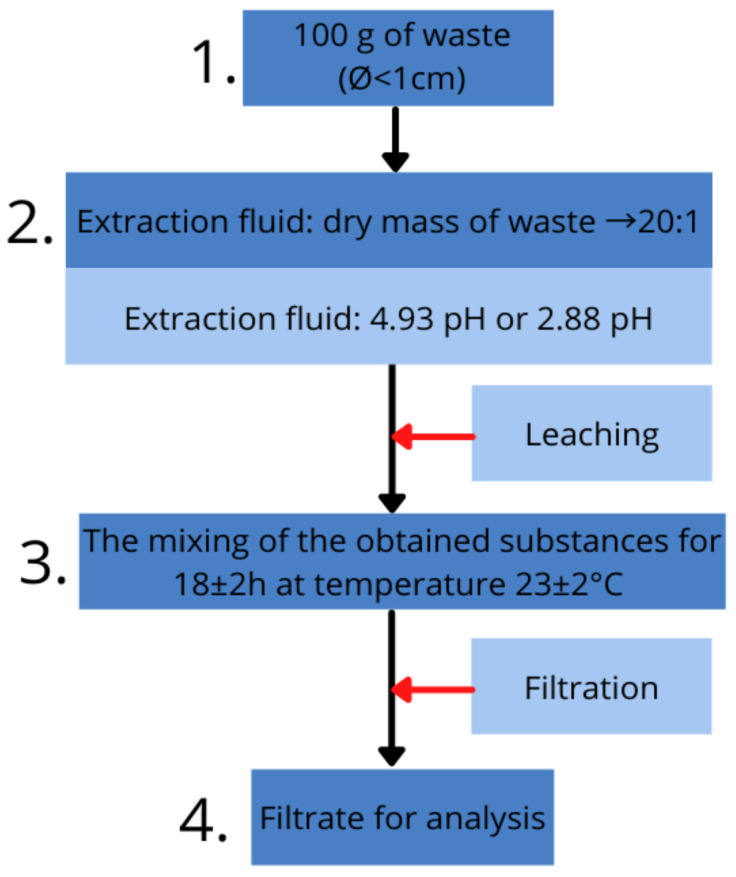
Diagram of TCLP test.

**Figure 5 materials-15-01001-f005:**
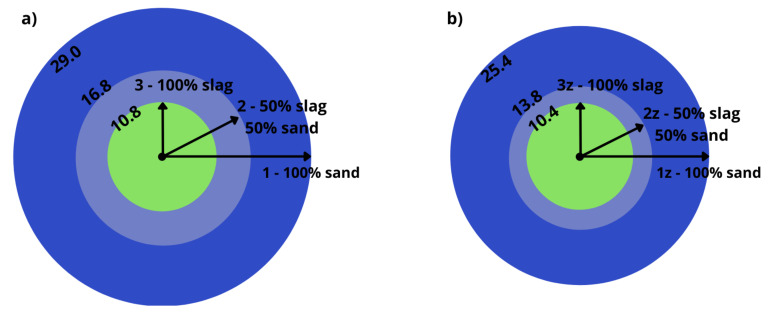
Consistency of fresh mortars based on sand and slag, with (**b**) and without (**a**) zeolite shown as the spreading diameter (cm) of the mortar.

**Figure 6 materials-15-01001-f006:**
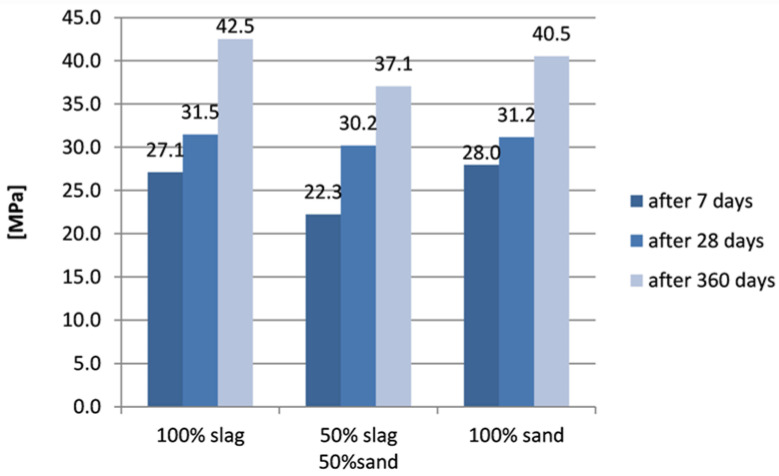
Compressive strength of samples without zeolite.

**Figure 7 materials-15-01001-f007:**
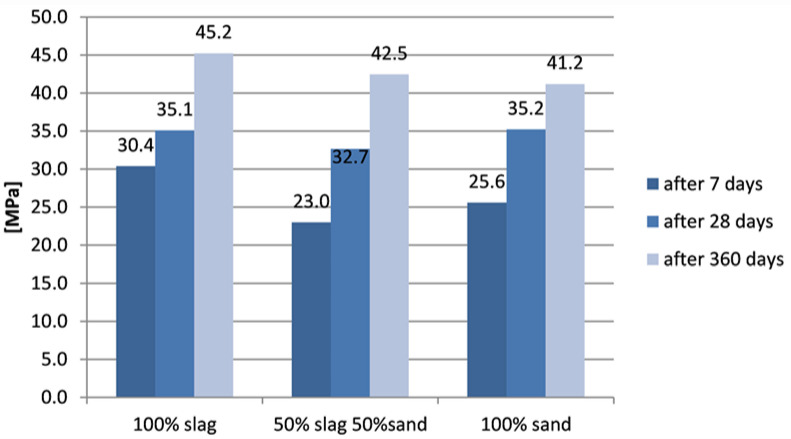
Compressive strength of samples with zeolite.

**Figure 8 materials-15-01001-f008:**
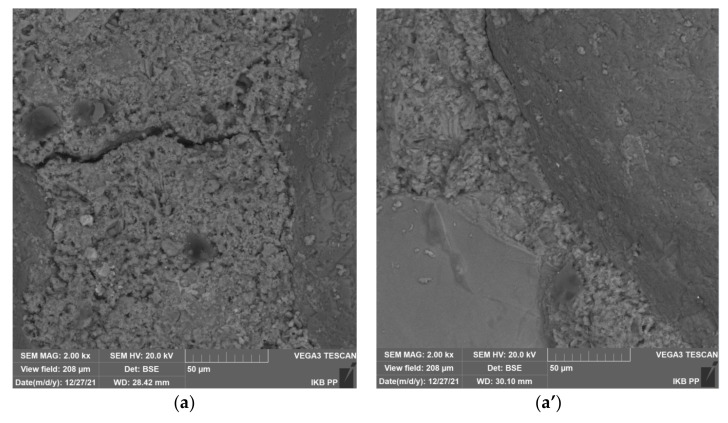
SEM images of cement mortars based on sand and sand with zeolite (**a**,**a’**), cement mortars based on slag and slag with zeolite (**b**,**b’**) and slag impurities (**c**,**c’**).

**Table 1 materials-15-01001-t001:** Oxide composition of cement, slag and zeolite.

Type of Sample	Oxide Composition, %											
	SiO_2_	K_2_O	CaO	Fe_2_O_3_	MnO	SrO	SO_3_	TiO_2_	CuO	Al_2_O_3_	ZnO	Cr_2_O_3_
Cement	17.6	0.9	67.2	3.9	0.2	0.13	3.8	0.28	-	4.3	-	-
Slag	18.0	2.1	39.8	21.7	0.52	0.46	4.6	2.2	0.65	7.5	1.73	0.27
Zeolite	60.9	8.89	9.73	7.91	0.13	0.23	-	0.82	0.07	11.0	-	-

**Table 2 materials-15-01001-t002:** The bulk density of sand and slag.

Aggregate	Bulk Density (kg/dm^3^)	
	Loose	Tapped
Slag	1.04	1.24
Sand	1.58	1.78

**Table 3 materials-15-01001-t003:** Flexural strength and density of samples after 7, 28, and 360 days.

Type of Sample	Flexural Strength (FS) and Density (D) after					
	7 Days		28 Days		360 Days	
	FS (MPa)	D (kg/m^3^)	FS (MPa)	D (kg/m^3^)	FS (MPa)	D (kg/m^3^)
1–100% slag	5.16	2109	5.50	2141	6.27	2109
1z–100% slag with zeolite	5.36	2113	5.96	2125	6.54	2246
2–50% slag and 50% sand	3.88	2184	5.03	2133	6.04	2176
2z–50% slag and 50% sand with zeolite	4.94	2207	5.15	2160	5.86	2230
3–100% sand	5.42	2176	5.74	2207	6.78	2234
3z–100% sand with zeolite	4.95	2184	6.60	2227	7.24	2277

**Table 4 materials-15-01001-t004:** Metal concentration in slag after leaching in different solutions.

Solution	Concentration of Leached Metals from Slag, mg/L					
	Cu	Pb	Ni	Zn	Fe	Cd
HCl	60.00	14.00	3.00	115.00	1556.00	<1.00
HNO_3_	85.00	14.00	3.00	111.00	700.00	<1.00
TCLP	2.60	2.00	0.00	15.00	5.00	1.44

**Table 5 materials-15-01001-t005:** Concentration of leached metals after 90 and 360 days.

Type of Sample	Concentration of Leached Metals from Cement Mortars, mg/L											
	Cu		Pb		Ni		Zn		Fe		Cd	
	90 Days	360 Days	90 Days	360 Days	90 Days	360 Days	90 Days	360 Days	90 Days	360 Days	90 Days	360 Days
1–100% slag	0	0	0	1	0	0	<0.5	0.9	0	1.0	0	0
1z–100% slag with zeolite	0	0	0	0	0	0	<0.5	0.5	0	0.5	0	0
2–50% slag and 50% sand	0	0	0	1	0	0	<0.5	0.6	0	0.5	0	0
2z–50% slag and 50% sand with zeolite	0	0	0	0	0	0	<0.5	0.6	0	0	0	0
3–100% sand	0	0	0	0	0	0	0	0.6	0	0	0	0
3z–100% sand with zeolite	0	0	0	0	0	0	0	0.6	0	0	0	0

**Table 6 materials-15-01001-t006:** Chloride and sulfate content in slag and cement mortars with different replacement of sand by MSWI slag.

Type of Sample	Sulfate Content (%)		Chloride Content (%)	
	After 90 Days	After 360 Days	After 90 Days	After 360 Days
Slag	1.70	1.67	0.25	0.14
1–100% slag	2.33	2.38	0.21	0.10
1z–100% slag with zeolite	2.41	1.85	0.18	0.06
2–50% slag and 50% sand	1.53	1.63	0.13	0.02
2z–50% slag and 50% sand with zeolite	1.96	1.52	0.12	0.03
3–100% sand	1.36	0.81	0.05	0.00
3z–100% sand with zeolite	1.04	0.89	0.05	0.02

## Data Availability

The data presented in this study are available on request from the corresponding author.

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
