# Peer review of "Long-Term Behavior of Cement Mortars Based on Municipal Solid Waste Slag and Natural Zeolite—A Comprehensive Physico-Mechanical, Structural and Chemical Assessment"

_materials, 2022, doi:10.3390/ma15031001_

Round 1

Reviewer 1 Report

The paper focuses on the long-term behavior of cement mortars based on municipal solid waste slag as aggregates and natural zeolite to partially replace Portland cement. The results are interesting. However, the paper should be revised before it desired to be published. The detailed suggestions are listed as follows.

1 The title should be more accurate according to the paper content. Please revise it.

2 In the abstract, the influence of zeolite seems not to be mentioned. Please confirm it.

3 The abbreviation MSWI should be given its full name when it appears for the first time. Please revise it.

4 It is more appropriate that the Section 3.1. Cement, zeolite and slag characteristics is set in 2.1 Materials, since the section mainly focus on the characteristics of materials used for cement mortar preparation. And the 2.2 was revised as cement mortar preparation.

5 In table 1, the component results of Portland cement may be wrong. Why the SiO2 content is so low, the CaO content is so high, and no Al2O3 is in the cement? Please confirm it.

6 In Fig.1, it is hard to distinguish the mineral phase related to the XRD peaks on the spectrum. It may be better to directly label the phase on the XRD peaks. Please confirm it.

7 Fig. 4 is hard to understand. Please revise it.

8 It may be more suitable to test the leaching concentration of Cr, since there is high Cr2O3 content in the slag. Please confirm it.

Author Response

Dear Reviewer, Thank You for Your insightful review of our work, which contributed to a better understanding of the scientific problems related to the subject of the publication and will help with the elimination of potential errors in the future.We would also like to express our gratitude for the revision of our manuscript and the opportunity to re-submit it, incorporating all of the Referees’ suggestions. Our comments and changes are noted below, and are marked in yellow in the manuscript. The paper focuses on the long-term behavior of cement mortars based on municipal solid waste slag as aggregates and natural zeolite to partially replace Portland cement. The results are interesting. However, the paper should be revised before it desired to be published. The detailed suggestions are listed as follows. Query 1: The title should be more accurate according to the paper content. Please revise it.Answer 1: We thank the Reviewer for this suggestion. A proposal for a new, more specific title “Long-term behavior of cement mortars based on municipal solid waste slag and natural zeolite - a comprehensive physico-mechanical, structural and chemical assessment ” was proposed in the paper and marked in yellow. Query 2: In the abstract, the influence of zeolite seems not to be mentioned. Please confirm it.Answer 2: We thank the Reviewer for this suggestion. Indeed, we did not include information about zeolite in the introduction; the characterization of the material is included in the materials section. Nevertheless, as requested by the reviewer, we shifted the information about zeolite to the introduction. Query 3: The abbreviation MSWI should be given its full name when it appears for the first time. Please revise it.Answer 3: We thank the Reviewer for this suggestion – we added the full name in the abstract. Query 4: It is more appropriate that the Section 3.1. Cement, zeolite and slag characteristics is set in 2.1 Materials, since the section mainly focus on the characteristics of materials used for cement mortar preparation. And the 2.2 was revised as cement mortar preparation.Answer 4: We thank the Reviewer for this suggestion – the Section 3.1 was moved to Section 2.1. Query 5: In table 1, the component results of Portland cement may be wrong. Why the SiO2 content is so low, the CaO content is so high, and no Al2O3 is in the cement? Please confirm it.Answer 5:  We thank the reviewer for this comment. Indeed, the data for cement placed in the Table 1 were incomplete. We have performed the XRF analysis again and the correct data has been posted. Query 6: In Fig.1, it is hard to distinguish the mineral phase related to the XRD peaks on the spectrum. It may be better to directly label the phase on the XRD peaks. Please confirm it.Answer 6: We thank the Reviewer for this suggestion. Placing phase descriptions directly above the peaks where different phases overlap will also, in our opinion, not significantly improve the readability of the XRD analysis. Therefore, we decided to add a table below the graph where we give specific values of the 2theta angle for each phase in a given material. Query 7: Fig. 4 is hard to understand. Please revise it.Answer 7: We thank the Reviewer for this suggestion – we added some explanations in the caption of the Fig.4 (now Fig.5). Query 8: It may be more suitable to test the leaching concentration of Cr, since there is high Cr2O3 content in the slag. Please confirm it.Answer 8: We thank the Reviewer for this query. We would like to explain that in eluates obtained for leaching trials performed for pure slag Cr was not detected. Therefore, Table 4 shows the metal concentrations that were present in the eluates. According to the Reviewer's suggestion and the literature and data from our waste incinerator, slag can contain harmful type elements. Zn, Cu, Pb, Ni, Cd, As and the Cr mentioned by the Reviewer. However, in a sample of the investigated slag and mortars made with it we have not recorded the presence of chromium, if it is present it is present in trace amounts, below the detection threshold. 

The whole manuscript has also been carefully checked with regard to editorial and language issues.

We look forward to hearing from you.

Yours faithfully,

Prof. Agnieszka Åšlosarczyk (corresponding author),

Poznan University of Technology

Reviewer 2 Report

The article contains a lot of valuable information on the topic presented. The content of the work does not raise major objections. The validity of the research topic is most appropriate. Minor revisions are recommended for the article to be published:

  1. In the introduction, the novelty of the present work, in relation to other thematically similar research works, should be clearly emphasized.
  2. Figure 1 should be slightly enlarged for better readability
  3. The axis designation is missing in figure 3 - please complete it.
  4. Conclusions should have more information regarding the quantitative evaluation of the research results.

Author Response

Dear Reviewer, Thank You for Your insightful review of our work, which contributed to a better understanding of the scientific problems related to the subject of the publication and will help with the elimination of potential errors in the future.We would also like to express our gratitude for the revision of our manuscript and the opportunity to re-submit it, incorporating all of the Referees’ suggestions. Our comments and changes are noted below, and are marked in yellow in the manuscript. The article contains a lot of valuable information on the topic presented. The content of the work does not raise major objections. The validity of the research topic is most appropriate. Minor revisions are recommended for the article to be published: Query 1: In the introduction, the novelty of the present work, in relation to other thematically similar research works, should be clearly emphasized.Answer 1: We thank the Reviewer for this suggestion – we added an explanation of the novelty of the research. Query 2: Figure 1 should be slightly enlarged for better readabilityAnswer 2: We thank the Reviewer for this suggestion. We modified Figure 1 by adding a table with the 2theta angle values for each phase to make it more readable. Query 3: The axis designation is missing in figure 3 - please complete it..Answer 3: We thank the Reviewer for this suggestion – axis description has been added. Query 4: Conclusions should have more information regarding the quantitative evaluation of the research results.Answer 4: We thank the Reviewer for this suggestion. The conclusions were supplemented with specific quantitative values as suggested by the Reviewer. 

The whole manuscript has also been carefully checked with regard to editorial and language issues.

We look forward to hearing from you.

Yours faithfully,

Prof. Agnieszka Åšlosarczyk (corresponding author),

Poznan University of Technology

Reviewer 3 Report

The paper by Marta Thomas, Małgorzata Osińska and Agnieszka Ślosarczyk presents an experimental study of cement mortars with municipal solid waste MSU slags. The use of MSU slag as aggregate in concrete pursues two objectives, one is to reduce the use of increasingly scarce natural aggregates and the second to get rid of a hazardous waste result from MSW incineration. The study focuses on the mechanical and lixiviate properties of cement mortar using slag with and without zeolite. The study concludes that mechanical properties are similar to the ones of mortars using natural sand as aggregate and heavy metal and other pollutants in lixiviates are not detectable or very low. Sulfates and chlorides experiment a slight increase with slag content.

Broad comments

The reviewer thanks the authors for the manuscript in a very interesting topic, some comments are given below:

  1. Abstract: in line 13, the jump from the first sentence to: Therefore, 3 types… is very sudden. Since the abstract is not very long, you can add some transition, for example describe what MSU slag is and why it is desirable to be recycled.
  2. To give more strength to the topic, give some numbers of percentage of incinerated waste in different countries and an estimation of total quantity of MSW slag produced.
  3. Reference 2 (Lam et al. 2011) is not about energy recovery from incineration even if it mentions it in the introduction, find a more precise reference.
  4. Line 38. If there are many studies you could cite a few.
  5. Introduction, especially the first paragraph, can be improved by extending it and improving English. For example, in first line, what is waste management hierarchy, some legal concept? Let the reader know without need to go to reference [1]. Or in line 29: energy is recovered… you can extend a bit explaining that heat from exhaust is used in a boiler which in turn runs a steam turbine…
  6. Lines 100-101: not clear what improved practice refers to
  7. Procedures described in Materials and Methods are fine, and since they are accompanied of the corresponding norm references they allow for reproducibility. Nevertheless, the reading through the section can be tedious. You can add some diagram or picture of the actual experimental set-up to ease understanding of the most complex ones.
  8. Lines 134-135: “loading rates 2mm/min rising to a max of 3000 N/m” is the test under displacement control or stress control? Or it is displacement controlled with a maximum in the stress rate?
  9. There is no explanation about Figure 4, which one is with zeolite?
  10. Line 347: what are the acceptable values?
  11. One of the conclusions is that the use of MSW slag increases sulfate and chloride content. Since these two components can cause steel corrosion, can you make some comments about the use of MSW slag in reinforced concrete (RC) for structural applications? How the values compare to limits established in the norms?

Specific comments

Line 178: a preliminary

Units in vertical axis of Figure 3

Units in vertical axis of Figure 5 and 6

Table 5 caption: leached metals from where? Cement mortars? Specify

Table 6 caption: specify more, leachate from mortar…

Sentence in lines 368-369: check grammar? Considering what is the cause and what the consequence.

Author Response

Dear Reviewer, Thank You for Your insightful review of our work, which contributed to a better understanding of the scientific problems related to the subject of the publication and will help with the elimination of potential errors in the future.We would also like to express our gratitude for the revision of our manuscript and the opportunity to re-submit it, incorporating all of the Referees’ suggestions. Our comments and changes are noted below, and are marked in yellow in the manuscript. The paper by Marta Thomas, MaÅ‚gorzata OsiÅ„ska and Agnieszka Åšlosarczyk presents an experimental study of cement mortars with municipal solid waste MSU slags. The use of MSU slag as aggregate in concrete pursues two objectives, one is to reduce the use of increasingly scarce natural aggregates and the second to get rid of a hazardous waste result from MSW incineration. The study focuses on the mechanical and lixiviate properties of cement mortar using slag with and without zeolite. The study concludes that mechanical properties are similar to the ones of mortars using natural sand as aggregate and heavy metal and other pollutants in lixiviates are not detectable or very low. Sulfates and chlorides experiment a slight increase with slag content.The reviewer thanks the authors for the manuscript in a very interesting topic, some comments are given below: Query 1: Abstract: in line 13, the jump from the first sentence to: Therefore, 3 types… is very sudden. Since the abstract is not very long, you can add some transition, for example describe what MSU slag is and why it is desirable to be recycled.Answer 1: We thank the Reviewer for this suggestion – we have added some explanations in the abstract. Query 2: To give more strength to the topic, give some numbers of percentage of incinerated waste in different countries and an estimation of total quantity of MSW slag produced.Answer 2: We thank the Reviewer for this suggestion. We supplemented the introduction with data on incinerator slag production using examples from available data and articles for Europe and Asian countries. Query 3: Reference 2 (Lam et al. 2011) is not about energy recovery from incineration even if it mentions it in the introduction, find a more precise.Answer 3: We thank the Reviewer for this suggestion - more precise reference is given. Query 4: Line 38. If there are many studies you could cite a few.Answer 4: We thank the Reviewer for this suggestion - a translation error appeared. The data we refer to, are the observations of the slag manufacturer who conducts his own research on its properties, but does not publish them anywhere. Nevertheless, we also cited literature sources. We changed this sentence to reflect the actual state of affairs: “According to the observations carried out by the manufacturer and some literature data, seasoning the slag in the air for 1 to 6 months increases the resistance to leaching.” Query 5: Introduction, especially the first paragraph, can be improved by extending it and improving English. For example, in first line, what is waste management hierarchy, some legal concept? Let the reader know without need to go to reference [1]. Or in line 29: energy is recovered… you can extend a bit explaining that heat from exhaust is used in a boiler which in turn runs a steam turbine…Answer 5: We thank the Reviewer for this suggestions - we included them in the introduction. Query 6: Lines 100-101: not clear what improved practice refers toAnswer 6: We thank the Reviewer for this suggestion - a translation error appeared. We have corrected this sentence to: “ Determination of consistency of fresh mortar was determined according to the standard [15].” Query 7: Procedures described in Materials and Methods are fine, and since they are accompanied of the corresponding norm references they allow for reproducibility. Nevertheless, the reading through the section can be tedious. You can add some diagram or picture of the actual experimental set-up to ease understanding of the most complex ones.Answer 7: We thank the Reviewer for this suggestion – we have added a diagram of TCLP test, and we have shortened some of the descriptions of study procedures. Query 8: Lines 134-135: “loading rates 2mm/min rising to a max of 3000 N/m” is the test under displacement control or stress control? Or it is displacement controlled with a maximum in the stress rate?Answer 8: We thank the Reviewer for this suggestion – the test is carried out under stress control.  Query 9: There is no explanation about Figure 4, which one is with zeolite?Answer 9: We thank the Reviewer for this suggestion – we added some explanations in the caption of the Fig.4 ( now Fig. 5). Query 10: Line 347: what are the acceptable values?Answer 10:  We thank the Reviewer for this query. We have changed the sentence from: ”However, the acceptable concentration was not exceeded for any of the tested samples and for any of the metals.” on: ”However, their concentration in leachate was less than permissible according to a TCLP test [20] and also concentration permissible for water eluates was not exceeded for any of the tested samples and for any of the metals.”  Query 11: One of the conclusions is that the use of MSW slag increases sulfate and chloride content. Since these two components can cause steel corrosion, can you make some comments about the use of MSW slag in reinforced concrete (RC) for structural applications? How the values compare to limits established in the norms?Answer 11: We thank the Reviewer for this query. Indeed, one of the problematic compounds in slag is the presence of chlorides and sulfates, the presence of which is not advisable in steel reinforced structures because of their potential to affect corrosion. Therefore, these quantities should be as low as possible. The amounts of sulfate determined in the paper are relatively low, less than 3%. These are the amounts of sulfate allowed in fly ash for concrete posted in the standard, for example. Nevertheless, their presence can affect the durability of the mortars and concretes themselves, both through the formation of expanding salts, which can lead to internal stresses, and, in the case of steel-reinforced structures, to corrosion of the reinforcement. The issues of structural durability are very important and clarifying and counteracting the adverse effects of municipal incinerator slag will contribute to its wider use in construction. In addition to these issues, the leachability of heavy metals over longer exploitation times is important, as well as hydrogen evolution due to the presence of aluminum in the slag, as pointed out by various researchers. The authors of this paper are aware of the disadvantages of this material, and therefore an ongoing study aims to investigate the effects of various external factors on the durability of mortars and concretes made from municipal waste incinerator slag. We thank you for the Reviewer's suggestion and will additionally include the issues of corrosion of reinforcement in the presence of sulfate and chloride in this research. 

Specific comments

Line 178: a preliminary

Units in vertical axis of Figure 3

Units in vertical axis of Figure 5 and 6

Table 5 caption: leached metals from where? Cement mortars? Specify

Table 6 caption: specify more, leachate from mortar… 

We thank the Reviewer for this suggestion – in the revised manuscript we have changed the caption for Tables 5 and 6, we have added units on Figure 3,5(6) and 6(7) and fixed a typing error in line 178.

Sentence in lines 368-369: check grammar? Considering what is the cause and what the consequence. „However, the chloride content is the higher, the higher the slag content in the sample”.

We thank the Reviewer for this suggestion – we reorganized this sentence and changes were marked in yellow.

The whole manuscript has also been carefully checked with regard to editorial and language issues.

We look forward to hearing from you.

Yours faithfully,

Prof. Agnieszka Åšlosarczyk (corresponding author),

Poznan University of Technology

Round 2

Reviewer 1 Report

The paper has been revised according to the reviewers‘ suggestions, so it may be published in the journal of materials .